# Vitreomacular Interface Disorders in Proliferative Diabetic Retinopathy: An Optical Coherence Tomography Study

**DOI:** 10.3390/jcm11123266

**Published:** 2022-06-07

**Authors:** Aidi Lin, Honghe Xia, Anlin Zhang, Xinyu Liu, Haoyu Chen

**Affiliations:** Joint Shantou International Eye Center, Shantou University and the Chinese University of Hong Kong, Shantou 515041, China; lad@jsiec.org (A.L.); xhh@jsiec.org (H.X.); zal@jsiec.org (A.Z.); liuxinyu@jsiec.org (X.L.)

**Keywords:** optical coherence tomography, prevalence, proliferative diabetic retinopathy, vitreomacular interface

## Abstract

Vitreomacular interface plays an important role in the pathogenesis and progression of proliferative diabetic retinopathy (PDR). This study investigated the prevalence and risk factors of vitreomacular interface disorders (VMID) in PDR. The macular optical coherence tomography (OCT) scans of 493 eyes from 378 PDR patients were retrospectively reviewed to detect VMID, including vitreomacular adhesion (VMA), vitreomacular traction (VMT), epiretinal membrane (ERM), lamellar hole–associated epiretinal proliferation (LHEP), and macular hole (MH). The associations between VMID and baseline factors, intraretinal structure, and visual acuity were analyzed. The prevalence was 78.9% for ERM, 13.4% for VMT, 4.8% for MH, 2.2% for LHEP, and 2.0% for VMA, respectively. On multivariable analyses (odds ratio, 95% confidence interval), fibrovascular proliferation (FVP) was positively associated with MH (8.029, 1.873–34.420), VMT (3.774, 1.827–7.798), and ERM (2.305, 1.460–3.640). High-risk PDR was another risk factor of ERM (1.846, 1.101–3.090). Female gender was positively associated with MH (3.836, 1.132–13.006), while vitreous hemorrhage was negatively associated with MH (0.344, 0.133–0.890). Eyes with all VMID subtypes showed more frequent macular cysts and tractional retinal detachment with poorer visual acuity (*p* ≤ 0.001). Therefore, the prevalence of VMID was considerably high, indicating that this distinct entity should be considered in interventions for PDR.

## 1. Introduction

Optical coherence tomography (OCT) is an essential tool to visualize the vitreomacular interface and detect vitreomacular interface disorders (VMID), including vitreomacular adhesion (VMA), vitreomacular traction (VMT), epiretinal membrane (ERM) and macular hole (MH) [1]. With aging, VMID can occur idiopathically and their pathophysiology is mainly based on vitreous liquefaction and incomplete posterior vitreous detachment [2]. They can also arise as complications of some ocular diseases, such as uveitis, retinitis pigmentosa, and macular edema [3,4,5]. Investigation of secondary VMID may help to understand the pathogenesis, management, and prognosis of these diseases.

Proliferative diabetic retinopathy (PDR) is characterized by neovascularization or preretinal/vitreous hemorrhage [6]. The metabolic factors in diabetes mellitus can cause the early liquefaction and cross-linking framework of the vitreous, leading to incomplete posterior vitreous detachment and vitreoschisis [7]. At the vitreomacular interface, tight adhesions exist between the vitreous and retinal blood vessels, where the posterior hyaloid acts as a scaffold for the growth of neovascular complexes [8]. The internal limiting membrane (ILM) is another scaffold that supports glial proliferation, probably inducing persistent macular edema and recurrent ERM [9]. Therefore, the vitreomacular interface in PDR is believed to be a complex plane involved in the pathogenesis and progression of the disease and as such deserves more attention. Although the clinical features and treatment outcomes of some VMID subgroups have been reported [10], the overall prevalence of VMID in PDR has not yet been investigated.

This study investigated the prevalence and associated factors of each type of VMID in PDR. The impacts on the intraretinal structure and visual acuity were also evaluated.

## 2. Materials and Methods

### 2.1. Study Subjects

This cross-sectional study obtained Institutional Board approval and followed the tenets of the Declaration of Helsinki. Because of the retrospective nature of the study, informed consent was waived. We searched the outpatient database of Joint Shantou International Eye Center of Shantou University and the Chinese University of Hong Kong between January 2014 and August 2019. Then we reviewed the records of patients with a diagnosis of PDR.

Inclusion criteria were as follows: (1) patients diagnosed as PDR with neovascularization or preretinal/vitreous hemorrhage in diabetic patients; (2) patients with macular OCT scan images in their first visit. The exclusion criteria were: (1) ocular comorbidities such as retinal vein occlusion, age-related macular degeneration, glaucoma, and pathologic myopia, defined as an excessive axial elongation related to myopia leading to structural changes in the posterior segment [11]; (2) OCT scans obtained only after vitrectomy; and (3) severe media opacity including vitreous hemorrhage and cataracts that made the OCT scans unreliable.

### 2.2. Data Collection

We collected the data from clinical records, including age, gender, study eye, prior history of hypertension, diabetic macular edema (DME), or ocular surgeries [cataract surgery/panretinal photocoagulation (PRP)/intravitreal injection of anti-vascular endothelial growth factor (VEGF) agents or triamcinolone acetonide]. Vitreous hemorrhage was recognized on ultrasonographic or fundus photographs. Fibrovascular proliferation (FVP) was defined as visible neovascular proliferation outside the macula based on fundus photographs. The PDR severity was classified as a “non-high-risk” or “high-risk” pattern according to the Early Treatment Diabetic Retinopathy Study. High-risk PDR was defined as follows: (1) new vessels on the disc (NVD) ≥ photograph 10A (about 1/4–1/3 disc area); or (2) vitreous/preretinal hemorrhage plus new vessels, either NVD < photograph 10A or new vessels elsewhere (NVE) ≥ 1/4 disc area [12]. The best-corrected visual acuity (BCVA) was converted to the unit of LogMAR for statistical analysis. A LogMAR of 2.0 was assigned to counting finger, 3.0 to hand motion, and 4.0 to light perception [13].

### 2.3. Evaluation of OCT Images

The macular cross-sectional cube scans were acquired from the following OCT devices: Cirrus HD-OCT 4000/5000 (Carl Zeiss Meditec, Dublin, CA, USA), Topcon 3D OCT-2000, or Topcon Triton (Topcon Corporation, Tokyo, Japan). Two ophthalmologists (A.L. and H.X.) who were masked for the clinical records examined the OCT images simultaneously. A third and senior retinal specialist (H.C.) made the final decision in case of disagreement.

The OCT images were graded for six types of VMID from three aspects (Figure 1): (1) vitreomacular interface relationship: VMA was defined as the perifoveal vitreous detachment from the retina but remaining attached within the fovea without retinal abnormalities; while those cases with retinal abnormalities due to traction from the adhered vitreous within the fovea were defined as VMT [1]; (2) epiretinal proliferation: ERM was defined as hyperreflective tissue along the retinal surface [14], while the thick and homogenous layer of material with medium reflectivity was considered as lamellar hole–associated epiretinal proliferation (LHEP), usually accompanied by retinal defects [15]; (3) macular hole (MH): Lamellar macular hole (LMH) was defined as a partial-thickness defect in the retinal layers with an irregular foveal contour, while full-thickness macular hole (FTMH) was characterized by a defect of all retinal layers from ILM to the retinal pigment epithelium [14].

The OCT images were also graded for the following intraretinal changes (Figure 2): (1) macular cysts were defined as the presence of hypo-reflective cysts in the retinal layer [16]; (2) retinoschisis was defined as lamellar splitting of the neurosensory retina, involving the inner or outer nuclear layer with the bridging columnar tissues frequently observed [17]; (3) retinal detachment was defined as a low-reflective space between the retinal pigment epithelium layer and the neurosensory retina; (4) tractional retinal detachment (TRD) was characterized by a relatively sharp border and concave curvature of the retina, while localized convex retinal detachment without traction was considered serous retinal detachment (SRD) [17]; (5) ellipsoid zone (EZ) disruption was defined as the incontiguous second hyper-reflective band of the outer retinal layers [18]; (6) disorganization of retinal inner layers (DRIL) was defined as the inability to delineate any of the boundaries of the ganglion cell layer–inner plexiform complex, inner nuclear layer, or outer plexiform layer [19]; (7) hard exudates were defined as hyperreflective dots in the neurosensory retina [20]; (8) the central macular thickness (CMT) was manually measured and defined as the distance from ILM to the retinal pigment epithelium at the fovea.

### 2.4. Statistical Analysis

Because the data was presented as categorical variables, the Kappa statistic was used to assess the interobserver concordance (A.L. and H.X.) in the OCT evaluation. Interpretations for Kappa were: ≤0, no agreement; 0.01–0.20, slight agreement; 0.21–0.40, fair agreement; 0.41–0.60, moderate agreement; 0.61–0.80, substantial agreement; and 0.81–1.00, almost perfect agreement [21]. The quantitative variables were reported as mean ± standard deviation (SD), and the categorical variables were expressed as No. (%). The prevalence of the VMID and its subtypes in PDR was calculated. We compared the baseline characteristics, intraretinal structure, and visual acuity in PDR patients with and without VMID [VMID (+) and VMID (−) group], as well as in MH eyes with and without LHEP. The independent-sample student *t*-test/Mann-Whitney U test for continuous variables and Chi-square/Fisher’s exact test for categorical variables were used. A *p*-value less than 0.05 was considered to be statistically significant. The associated factors in VMID were investigated using univariate and multivariable logistic regression analyses. In the univariate analyses, factors with *p* < 0.05 were included in the multivariable analyses. Moreover, generalized estimating equations were used to account for interocular correlation. Associations were presented as odds ratio (OR) with a 95% confidence interval (CI). Statistical analyses were performed by SPSS software version 19 (SPSS, Inc., Chicago, IL, USA).

## 3. Results

### 3.1. Background Characteristics of Patients

In this study, 516 PDR eyes were approached, but 23 eyes were ineligible based on the exclusion criterion. In the end we included 493 eyes from 378 PDR patients. These consisted of 243 right and 250 left eyes from 137 males and 241 females. The age was 55.6 ± 8.4 years old. The interobserver concordance in the OCT evaluation was almost perfect, with the Kappa values ranging from 0.901 to 0.999 (Appendix A).

### 3.2. Prevalence of VMID in PDR

Table 1 shows that the overall prevalence of VMID was 81.3% (401/493 eyes). In terms of the vitreomacular relationship, the prevalence of VMT was 13.4%, while that of VMA was only 2.0%. For the epiretinal proliferation, the prevalence of ERM was considerably high (78.9%), while that of LHEP was only 2.2%. Meanwhile, all LHEP cases were concurrent with ERM. The prevalence of MH was 4.8% (3.0% for LMH and 1.8% for FTMH).

### 3.3. Factors Associated with VMID in PDR

Table 2 compares baseline factors in PDR with and without VMID. FVP and high-risk PDR were more common in all VMID subtypes than in the VMID (−) group (*p* ≤ 0.012). A lower presence rate of vitreous hemorrhage was observed in LMH than in the VMID (−) group (20% vs. 51%, *p* = 0.025). However, other factors showed no differences in groups with and without VMID (*p* > 0.508).

Table 3 shows the univariate logistic regression analyses of the factors associated with VMID in PDR. In the univariate model, both FVP (OR: 4.297, 95% CI: 2.142–8.621, *p* < 0.001) and high-risk PDR (OR: 3.121, 95% CI: 1.281–7.603, *p* = 0.012) were positively associated with VMT. However, in the multivariate model, only FVP was proved to be a risk factor of VMT (OR: 3.774, 95% CI: 1.827–7.798, *p* < 0.001). FVP (OR: 2.305, 95% CI: 1.460–3.640, *p* < 0.001) and high-risk PDR (OR: 1.846, 95% CI: 1.101–3.090, *p* = 0.020) were both positively associated with ERM. Female gender (OR: 3.836, 95% CI: 1.132–13.006, *p* = 0.031) and FVP (OR: 8.029, 95% CI: 1.873–34.420, *p* = 0.005) were the risk factors of MH, while vitreous hemorrhage (OR: 0.344, 95% CI: 0.133–0.890, *p* = 0.028) was the protective factor.

### 3.4. Comparisons of Intraretinal Structure and Visual Acuity in PDR with and without VMID

Table 4 presents the intraretinal structure and visual acuity comparisons in PDR with and without VMID. Compared with the VMID (−) group, all VMID groups showed higher rates of macular cysts (*p* ≤ 0.006) and TRD (*p* < 0.001) with poorer visual acuity (*p* ≤ 0.001). Most VMID groups had more frequent retinoschisis except for FTMH (*p* = 0.982) and showed higher CMT except for LMH (*p* = 0.051). Only VMT showed a lower presence rate of SRD (0% vs. 5%, *p* = 0.019), while only FTMH presented a higher rate of EZ disruption (100% vs. 74%, *p* = 0.023).

### 3.5. Comparisons of the MH Eyes with and without VMID

In MH eyes, there were 10 cases of LHEP (6 in LHM eyes and 4 in FTMH eyes). The baseline factors, intraretinal structural appearances, and visual acuity showed no significant differences in groups with and without LHEP (*p* ≥ 0.060).

## 4. Discussion

To our knowledge, the prevalence of VMID in PDR patients has not yet been reported. In our study, VMID was quite prevalent in PDR and positively associated with FVP, presenting with more severe structural and visual impairments.

In PDR, the clinical and histopathological characteristics of ERM have been reported [10], but few researchers have investigated the prevalence of ERM before vitrectomy. Even when PDR patients have undergone vitrectomy, the recurrence rate of ERM was still high, ranging from 21.7% to 52.8% [9,16,20]. This proliferation was complex and vascularized, progressed rapidly, and induced macular distortion [10]. Our study showed a higher prevalence rate of ERM in PDR (78.9%) than in older adults (34.1%) [14]. The Maastricht Study has reported the prevalence of ERM in populations with different glucose metabolism statuses, only ranging from 5.7% to 6.7% [2]. ERM in DME was also common, with a presence rate ranging from 67% to 100% that varied in different types of DME [22]. This suggests that the common pathogenesis of DME and PDR may promote the development of ERM.

Our study showed that the prevalence of VMT was only secondary to ERM (13.4%), which was higher than in the general population (2.4% in the Beijing Eye Study, 1.6% in the Beaver Dam Eye Study, and 1.56% in a Belgian study cohort), [14,23,24]. The Maastricht Study showed a prevalence of 6% to 9.6% in participants with different glucose metabolism statuses [2]. A lower presence rate (11.7%) in PDR patients has been reported, but such reports only included cases of V-shaped VMT [16].

The prevalence of LMH in PDR patients was 3.0%, comparable to that of idiopathic LMH in older adults (3.6%) [14]. It has been shown that type 2 diabetes was negatively associated with LMH (OR = 0.2, *p* = 0.036) with a low prevalence rate of 0.3%, and women were more predisposed [2]. In contrast, the prevalence of FTMH in PDR was 1.8%, which was higher than that in older adults [14]. Our study demonstrated that female gender (OR = 3.836, *p* = 0.031) was the risk factor for MH in PDR patients. Moreover, MH with FVP in PDR patients has been regarded as a distinct entity with remarkable proliferative and tractional features [25]. The association between FVP and MH (OR = 8.029, *p* = 0.005) also indicated the prominent role of proliferation in MH formation. Fresh vitreous hemorrhage was an active PDR stage that may exert weaker tractional forces on the retina than the fibrotic stage [10], which probably explains the negative correlation with MH (OR = 0.344, *p* = 0.028).

LHEP was the preretinal tissue with homogenous medium reflectivity in many patients with VMID, especially LMH. The classification of the idiopathic LMH has been proposed, and the presence of LHEP and ERM/VMT indicated degenerative and tractional types, respectively [26]. In our PDR case series with MH, all LHEP eyes were concurrent with ERM. There was no evidence of any degenerative feature in PDR with LHEP. Therefore, this classification of idiopathic LMH was not adequately applied to MH in PDR.

The presence rates of hard exudates were not significantly different for all subtypes (*p* ≥ 0.122), and a lower rate of SRD was only observed in VMT (*p* = 0.019). However, all VMID subtypes showed higher presence rates of TRD (*p* < 0.001) with worse visual acuity (*p* ≤ 0.001). In PDR, traction may have a more deleterious effect on the intraretinal morphology than exudation, which explains the poor visual acuity in VMID eyes. Tractional and exudative factors can contribute to the higher rates of macular cysts in all VMID subtypes (*p* ≤ 0.006). As a significant indicator of macular elevation induced by FVP [17], retinoschisis was more common in most subtypes (*p* ≤ 0.013) except for FTMH (*p* = 0.982) in PDR. EZ disruption has been reported as a significant predictor of visual prognosis [27]. However, a higher presence rate was only shown in FTMH (*p* = 0.023), possibly suggesting a poorer prognosis in this subtype.

It has been reported that VMID appeared more common in DME patients who had undergone PRP and cataract surgery [28]. The annual incidence of VMID in DME eyes treated with anti-VEGF was 6.43% [29]. Nevertheless, VMID showed no associations with prior history of DME or intravitreal injection.

Although the pathogenesis of VMID in PDR has not been fully understood yet, an investigation of the risk factors makes sense. Our study showed the prominent role of FVP in the formation of VMID in PDR. With such a high prevalence rate, the status of the vitreomacular interface is believed to be significant and deserves early evaluation and routine monitoring.

Identifying VMID will enable interventions designed to prevent further deterioration in vision before irreversible retinal pathology occurs. Preoperative recognition of VMID would help clear the FVP and relieve the traction during the surgery. Furthermore, identifying the specific type is necessary to make a treatment decision. For example, observation is preferred for VMA, while vitrectomy is more appropriate for VMT and MH, the latter of which sometimes requires ILM peeling. Further studies are needed to investigate surgical outcomes for these types of patients.

Although intravitreal injection of anti-VEGF has been proved to prevent progression and even promote regression of PDR, it is off-label in China and cannot be reimbursed. Moreover, we included the first OCT scans of PDR patients. Thus, most of them were at the time of diagnosis or shortly after diagnosis. These may explain why the number of cases with prior intravitreal injection was low in this study.

Because this was a retrospective study, the macular scans were collected from different OCT devices that have been available in clinical practice. Notably, they have proved reliable enough to detect VMID [3,4,5,14,24,30,31]. Different OCT devices showed the advantages of making the conclusions more generalized and adaptive to various conditions. Regarding drawbacks, there might exist differences between devices for quantitative measurements. In this case, we measured CMT manually to reduce the risk of bias.

There are some limitations to our study. Firstly, in this hospital-based study, we may have overestimated prevalence. Secondly, those cases with severe medium opacity were excluded, which may cause a selection bias. Thirdly, the retrospective design entails certain drawbacks: The first visit with an OCT scan was included, and the time from diagnosis to inclusion may vary in different patients; the macular OCT scans were limited in the evaluation of VMID without multimodal and widefield imaging [32]; glucose level and types and duration of diabetes have not been collected because of the lack of records; and finally, the grading of VMID and intraretinal changes needs further investigation.

## 5. Conclusions

In conclusion, the prevalence of VMID was considerably high in PDR, and FVP was identified as the risk factor. PDR complicated with VMID showed more severe damage to the intraretinal structure and visual acuity. Therefore, VMID should be considered a distinct entity, detection of which using OCT is essential in interventions on PDR patients.

## Figures and Tables

**Figure 1 jcm-11-03266-f001:**
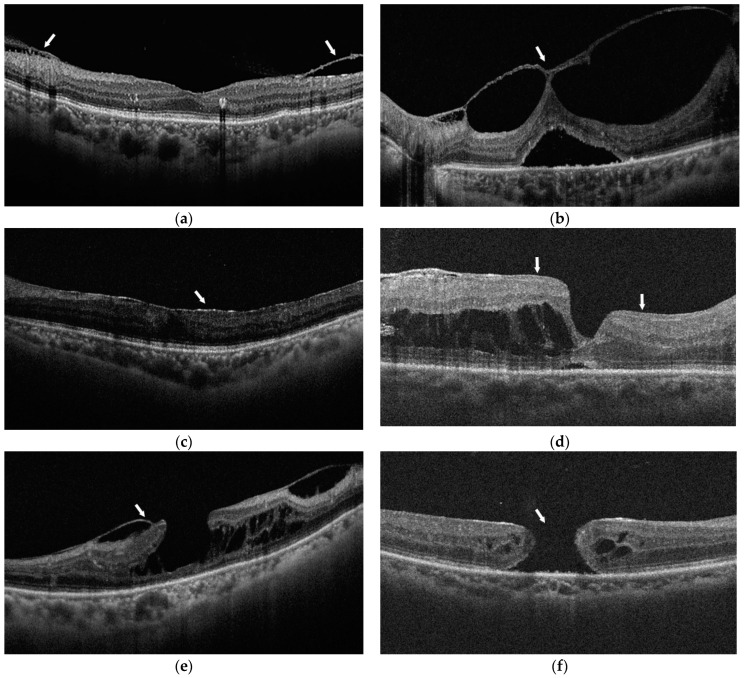
Classifications of vitreomacular interface disorders (VMID) (white arrows) on optical coherence tomography (OCT): (**a**) vitreomacular adherence (VMA); (**b**) vitreomacular traction (VMT); (**c**) epiretinal membrane (ERM); (**d**) lamellar hole–associated epiretinal proliferation (LHEP); (**e**) lamellar macular hole (LMH); (**f**) full-thickness macular hole (FTMH).

**Figure 2 jcm-11-03266-f002:**
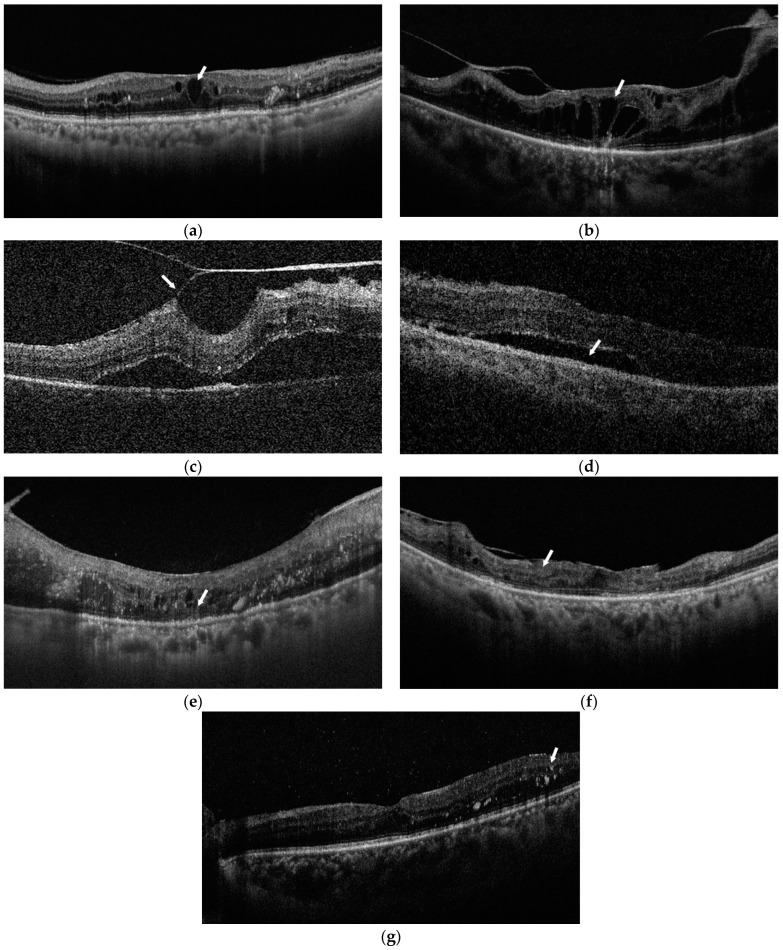
Categories of intraretinal changes (white arrows) on optical coherence tomography (OCT): (**a**) macular cysts; (**b**) retinoschisis; (**c**) tractional retinal detachment (TRD); (**d**) serous retinal detachment (SRD); (**e**) ellipsoid zone (EZ) disruption; (**f**) disorganization of retinal inner layers (DRIL); (**g**) hard exudates.

**Table 1 jcm-11-03266-t001:** Prevalence of vitreomacular interface disorders in proliferative diabetic retinopathy.

Type		N	% of VMID	% of PDR Sample
VMID		401	100	81.3
Vitreomacular relationship	VMA	10	2.5	2.0
	VMT	66	16.5	13.4
Epiretinal proliferation	ERM	389	97.0	78.9
	LHEP	11	2.7	2.2
Macular hole	LMH	15	3.7	3.0
	FTMH	9	2.2	1.8

Abbreviations: VMID, Vitreomacular interface disorders; N, numbers; PDR, proliferative diabetic retinopathy; VMA, vitreomacular adhesion; VMT, vitreomacular traction; ERM, epiretinal membrane; LHEP, lamellar hole–associated epiretinal proliferation; LMH, lamellar macular hole; FTMH, full-thickness macular hole.

**Table 2 jcm-11-03266-t002:** Baseline characteristics of proliferative diabetic retinopathy.

Characteristics	VMID (−) (N = 92)	VMID (+) (N = 401)	*p*	VMT (N = 66)	*p*	ERM (N = 389)	*p*	LMH (N = 15)	*p*	FTMH (N = 9)	*p*
Age, mean ± SD, year	56.9 ± 7.6	55.3 ± 8.6	0.186	55.0 ± 8.7	0.364	55.3 ± 8.6	0.192	55.9 ± 9.1	0.746	58.3 ± 4.2	0.508
Female, N (%)	57(62)	257(64)	0.701	44(67)	0.543	249(64)	0.713	13(87)	0.062	8(89)	0.080
Laterality of eye, right, N (%)	50(54)	193(48)	0.282	31(47)	0.360	183(47)	0.207	8(53)	0.942	6(67)	0.478
Hypertension, N (%)	51(55)	254(63)	0.159	44(67)	0.155	246(63)	0.166	8(53)	0.879	4(44)	0.527
Diabetic macular edema, N (%)	3(3)	11(3)	0.791	2(3)	0.935	10(3)	0.720	0(0)	0.478	0(0)	0.582
Prior cataract surgery, N (%)	6(7)	50(12)	0.105	10(15)	0.076	48(12)	0.112	2(13)	0.390	0(0)	0.282
Vitreous hemorrhage, N (%)	47(51)	203(51)	0.936	34(52)	0.958	200(51)	0.955	3(20)	**0.025**	3(33)	0.305
Prior intravitreal injection, N (%)	5(5)	20(5)	0.861	4(6)	0.867	19(5)	0.829	0(0)	0.213	0(0)	0.328
Prior PRP, N (%)	45(49)	194(48)	0.926	33(50)	0.893	186(48)	0.850	11(73)	0.079	3(33)	0.367
FVP, N (%)	34(37)	257(64)	**<0.001**	56(85)	**<0.001**	250(64)	**<0.001**	14(93)	**<0.001**	7(78)	**0.002**
High-risk PDR, N (%)	59(64)	331(83)	**<0.001**	61(92)	**<0.001**	322(83)	**<0.001**	14(93)	**0.012**	9(100)	**0.006**

Abbreviations: SD, standard deviation; N, numbers; PRP, panretinal photocoagulation; FVP, fibrovascular proliferation; PDR, proliferative diabetic retinopathy; VMID, vitreomacular interface disorders; VMID (+), the group with VMID; VMID (−), the group without VMID; VMT, vitreomacular traction; ERM, epiretinal membrane; LMH, lamellar macular hole; FTMH, full-thickness macular hole; *p*, comparing with VMID (−) group; *p*-values marked in bold indicate significance.

**Table 3 jcm-11-03266-t003:** Univariate analyses of the baseline factors associated with vitreomacular interface disorders in proliferative diabetic retinopathy.

Characteristics	VMT (N = 66)		ERM (N = 389)		MH (N = 24)	
	OR (95% CI)	*p*	OR (95% CI)	*p*	OR (95% CI)	*p*
Age	0.994 (0.962–1.027)	0.720	0.979 (0.956–1.000)	0.090	1.019 (0.973–1.067)	0.421
Female (vs. male)	1.172 (0.646–2.128)	0.600	1.070 (0.686–1.680)	0.760	4.173(1.239–14.063)	**0.021**
Hypertension	1.221 (0.683–2.183)	0.500	1.320 (0.845–2.050)	0.220	0.615 (0.273–1.385)	0.240
Diabetic macular edema	1.373 (0.351–5.368)	0.650	0.665 (0.207–2.140)	0.490	NA	NA
Prior cataract surgery	1.742 (0.829–3.659)	0.140	1.710 (0.769–3.790)	0.190	0.661 (0.145–3.018)	0.590
Vitreous hemorrhage	1.012 (0.616–1.663)	0.960	1.140 (0.735–1.770)	0.560	0.315 (0.124–0.799)	**0.015**
Prior intravitreal injection	1.465 (0.551–3.896)	0.440	0.837 (0.328–2.130)	0.710	NA	NA
Prior PRP	1.198 (0.674–2.131)	0.540	0.881 (0.574–1.350)	0.560	1.454 (0.639–3.309)	0.370
FVP	4.297 (2.142–8.621)	**<0.001**	2.770 (1.790–4.280)	**<0.001**	8.170 (1.926–34.687)	**0.004**
High-risk PDR	3.121 (1.281–7.603)	**0.012**	2.580 (1.580–4.210)	**<0.001**	6.312 (0.788–50.536)	0.083

Abbreviations: N, numbers; VMT, vitreomacular traction; ERM, epiretinal membrane; MH, macular hole; PRP, panretinal photocoagulation; FVP, Fibrovascular proliferation; PDR, proliferative diabetic retinopathy; OR, odds ratio; 95% CI, 95% confidence interval; NA, not applicable; *p*-values marked in bold indicate significance.

**Table 4 jcm-11-03266-t004:** Intraretinal structure and visual acuity in proliferative diabetic retinopathy.

Characteristics	VMID (−) (N = 92)	VMID (+) (N = 401)	*p*	VMT (N = 66)	*p*	ERM (N = 389)	*p*	LMH (N = 15)	*p*	FTMH (N = 9)	*p*
Macular cysts, N (%)	20(22)	168(42)	**<0.001**	31(47)	**0.001**	163(42)	**<0.001**	9(60)	**0.004**	6(67)	**0.006**
Retinoschisis, N (%)	10(11)	89(22)	**0.014**	20(30)	**0.002**	87(22)	**0.013**	6(40)	**0.002**	1(11)	0.982
TRD, N (%)	12(13)	164(41)	**<0.001**	54(82)	**<0.001**	157(40)	**<0.001**	11(73)	**<0.001**	9(100)	**<0.001**
SRD, N (%)	5(5)	17(4)	0.625	0(0)	**0.019**	17(4)	0.667	0(0)	0.213	0(0)	0.328
EZ disruption, N (%)	68(74)	327(82)	0.098	52(79)	0.480	318(82)	0.090	11(75)	0.962	9(100)	**0.023**
DRIL, N (%)	54(59)	242(60)	0.770	39(59)	0.960	235(60)	0.763	9(60)	0.924	7(78)	0.247
Hard exudates, N (%)	80(87)	329(82)	0.258	54(82)	0.375	317(81)	0.214	12(80)	0.490	9(100)	0.122
CMT (um)	236.7 ± 141.2	328.0 ± 206.5	**<0.001**	385.6 ± 262.1	**<0.001**	323.1 ± 197.8	**<0.001**	285.1 ± 358.8	0.051	610.6 ± 325.1	**<0.001**
BCVA, mean ± SD	0.83 ± 0.52	1.08 ± 0.69	**0.001**	1.10 ± 0.56	**0.001**	1.10 ± 0.69	**<0.001**	1.53 ± 0.87	**0.001**	1.63 ± 0.70	**0.001**

Abbreviations: N, numbers; SD, standard deviation; VMID, vitreomacular interface disorders; VMID (+), the group with VMID; VMID (−), the group without VMID; VMT, vitreomacular traction; ERM, epiretinal membrane; LMH, lamellar macular hole; FTMH, full-thickness macular hole; TRD, tractional retinal detachment; SRD, serous retinal detachment; EZ, ellipsoid zone; DRIL, disorganization of retinal inner layers; CMT, central macular thickness; um, micrometer; BCVA, best-corrected visual acuity (LogMAR); *p*, comparing with VMID (−) group; *p*-values marked in bold indicate significance.

## Data Availability

The data presented in this study are available on request from the corresponding author.

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
