# Peer review of "Vitreomacular Interface Disorders in Proliferative Diabetic Retinopathy: An Optical Coherence Tomography Study"

_jcm, 2022, doi:10.3390/jcm11123266_

Round 1

Reviewer 1 Report

I would like to congratulate the authors on their work. It is a very interesting observational study on the vitreoretinal interface changes encountered in patients with PDR.

My main concern is the need for a review by a native speaker as the there are some sentences that need to be rewritten to make them clearer. As there is an extensive results section with much data, this would greatly improve the clarity of the manuscript.

Regarding the structure of the paper, I have no major concerns.

Author Response

Point 1: My main concern is the need for a review by a native speaker as the there are some sentences that need to be rewritten to make them clearer. As there is an extensive results section with much data, this would greatly improve the clarity of the manuscript.

Response 1: Thanks for the precious suggestions. We have followed the advice and revised the manuscript to improve the clarity.

Reviewer 2 Report

This study investigated the prevalence and risk factors of vitreomacular interface disorders (VMID) in PDR. The main conclusion is that the prevalence of VMID was considerably high in PDR, and FVP was identified as the risk factor. PDR complicated with VMID showed more severe damage to both intraretinal structure and visual acuity. The topic of the paper is of clinical significance and may help and guide clinicians in everyday practice.

The study is of a retrospective nature. It also does not reflect the true prevalence of VMID in PDR, but the authors discuss that issue in the part “limitations of the study”.

Another important aspect of the studied population is the percentage of patients treated previously with anti-VEGF. Although intravitreal injection of anti-VEGF has been proved to prevent progression and even promote regression of PDR, it is off-label in China and cannot be reimbursed. Therefore, the studied population in China does not reflect the actual status in different countries.  

I have significant concerns regarding the proper matching of the examples of patients to the appropriate definition according to the classification of vitreomacular interface disorders (VMID). Figure 1, which is supposed to show and support the choice of the assignment of patients to the definitions, such as VMA, VMT, ERM), LHEP, LMH, FTMH), presents two questionable descriptions:

“(d) lamellar hole-associated epiretinal proliferation (LHEP)”; it seems to me, that the scan shows retinal detachment caused by epiretinal membrane and traction and associated with full-thickness macular hole.

“(f) full-thickness macular hole (FTMH)”; this is actually an extraordinary example of an FTMH associated with epiretinal traction and partial, central retinal detachment.

I would suggest replacing the scans in order to present classic examples of presented disorders. Also, check if this does not interfere with the final results you presented.

All parts of the manuscript are presented clearly and understandable without any significant English language errors. The methodology is clear and correct, except for one minor issue - please define “pathologic myopia”.

The tables are properly arranged and the table legends are clear for the reader.

The strength of this work is the practical solution for ophthalmologists treating patients with PDR.

Author Response

Point 1: I have significant concerns regarding the proper matching of the examples of patients to the appropriate definition according to the classification of vitreomacular interface disorders (VMID). Figure 1, which is supposed to show and support the choice of the assignment of patients to the definitions, such as VMA, VMT, ERM), LHEP, LMH, FTMH), presents two questionable descriptions:

“(d) lamellar hole-associated epiretinal proliferation (LHEP)”; it seems to me, that the scan shows retinal detachment caused by epiretinal membrane and traction and associated with full-thickness macular hole.

“(f) full-thickness macular hole (FTMH)”; this is actually an extraordinary example of an FTMH associated with epiretinal traction and partial, central retinal detachment.

I would suggest replacing the scans in order to present classic examples of presented disorders. Also, check if this does not interfere with the final results you presented.

Response 1: Thanks for the precious advice. We have replaced the figures to present classic examples of LHEP (figure 1d) and FTMH (figure 1f), respectively. And this does not interfere with the final results.

Point 2: The methodology is clear and correct, except for one minor issue - please define “pathologic myopia”.

Response 2: Thanks for the comments. The definition of “pathologic myopia” has been added in lines 68-69.

Reviewer 3 Report

In general, this study is very interesting and timely! I strictly recommend publication of this manuscript after a thorough overwork! Here are some comments and recommendations for improvement of the manuscript:

Title should be reformulated as it is too general maybe add "an OCT study"?

Line 81 cirrus model should be specified

Why did you use K coeficiency and not intercalss corelation coeficiency, test-retest repeatabilit and = within-subject coefficient of variation? Usually in measurring inter observer coeficiency these three parameters are used. 

Why were different OCT devices used? Please mention the benefits AND DRAWBACKS of each device for this study. 

Author Response

Point 1: Title should be reformulated as it is too general maybe add "an OCT study"?

Response 1: Thanks for the precious advice. We have revised the title to make it more specific (line 3).

Point 2: Line 81 cirrus model should be specified

Response 2: Thanks for the advice. We have added the model of Cirrus OCT in line 85.

Point 3: Why did you use K coeficiency and not intercalss corelation coeficiency, test-retest repeatabilit and = within-subject coefficient of variation? Usually in measurring inter observer coeficiency these three parameters are used. 

Response 3: Thanks for your kind suggestions. Because the data was presented as categorical variables, we used the Kappa statistic instead of the interclass correlation coefficient and within-subject coefficient of variation. The OCT images were evaluated by two graders simultaneously, thus the test-retest repeatability may not fit in this case. We are so sorry that we have not clarified it previously. And the expression has been revised in the manuscript. (lines 86-88; lines 130-132)

Point 4: Why were different OCT devices used? Please mention the benefits AND DRAWBACKS of each device for this study. 

Response 4: Thanks for the suggestions. Because this was a retrospective study, we obtained the macular images from different OCT devices available in clinical practice. Besides, all of them have been widely applied in evaluating the VMID. Different OCT devices showed the benefits of making the conclusions more generalized and adaptive in various conditions. Regarding drawbacks, there might exist differences between devices for quantitative measurements. In our study, we measured CMT manually to reduce the risk of bias. The manuscript has been revised. (lines 323-328)

Round 2

Reviewer 2 Report

In my review, I raised significant concerns regarding the proper matching of the examples of patients to the appropriate definition according to the classification of vitreomacular interface disorders (VMID). Figure 1, which was supposed to show and support the choice of the assignment of patients to the definitions, such as VMA, VMT, ERM), LHEP, LMH, FTMH), presented two improper descriptions:

“(d) lamellar hole-associated epiretinal proliferation (LHEP)”;

“(f) full-thickness macular hole (FTMH)”;

I suggested replacing the scans to present classic examples of presented disorders and also suggested checking if this did not interfere with the final results presented.

The authors reassured me, that they replaced the figures and this did not change and interfered with the final results.

The new figure 1(d) described: “lamellar hole-associated epiretinal proliferation (LHEP)” is a full-thickness MH.
This proves the lack of knowledge of the basic assignment of the definitions to the diagnosis of the state of the retina, and thus the unpredictable falsification of the obtained results.

The main purpose of the study was the investigation of the prevalence and risk factors of vitreomacular interface disorders (VMID) in PDR. It cannot be fulfilled without the appropriate assignment of the retinal state to the proper definition of the VMID. I understood it once and gave the authors the possibility to change the figures, but they replaced it with another wrongly assigned one.

Author Response

Response to Reviewer 2 Comments

Point 1: The new figure 1(d) described: “lamellar hole-associated epiretinal proliferation (LHEP)” is a full-thickness MH.
This proves the lack of knowledge of the basic assignment of the definitions to the diagnosis of the state of the retina, and thus the unpredictable falsification of the obtained results.

The main purpose of the study was the investigation of the prevalence and risk factors of vitreomacular interface disorders (VMID) in PDR. It cannot be fulfilled without the appropriate assignment of the retinal state to the proper definition of the VMID. I understood it once and gave the authors the possibility to change the figures, but they replaced it with another wrongly assigned one.

Response 1: Thanks for the comments. The definition of lamellar hole-associated epiretinal proliferation (LHEP) was not limited to the lamellar macular hole (LMH). Several articles have reported LHEP in the full-thickness macular hole (FTMH) [1-4]. Pang et al. demonstrated that 88.2% of the LHEP had LMH, while 11.8% had FTMH [5]. Therefore, these lesions are always concurrent with each other instead of occurring independently. The new figure 1(d) showed the LHEP (white arrows) around the edge of FTMH, which we considered a classic example to fit in the definition of LHEP.

References

  1. Choi, W.S.; Merlau, D.J.; Chang, S. VITRECTOMY FOR MACULAR DISORDERS ASSOCIATED WITH LAMELLAR MACULAR HOLE EPIRETINAL PROLIFERATION. Retina 2018, 38, 664-669, doi:10.1097/iae.0000000000001591.
  2. Lai, T.T.; Chen, S.N.; Yang, C.M. Epiretinal proliferation in lamellar macular holes and full-thickness macular holes: clinical and surgical findings. Graefes Arch Clin Exp Ophthalmol 2016, 254, 629-638, doi:10.1007/s00417-015-3133-9.
  3. Lai, T.T.; Yang, C.M. LAMELLAR HOLE-ASSOCIATED EPIRETINAL PROLIFERATION IN LAMELLAR MACULAR HOLE AND FULL-THICKNESS MACULAR HOLE IN HIGH MYOPIA. Retina 2018, 38, 1316-1323, doi:10.1097/iae.0000000000001708.
  4. Liu, J.; Lyu, J.; Zhang, X.; Zhao, P. Lamellar hole-associated epiretinal membrane is a common feature of macular holes in retinitis pigmentosa. Eye (Lond) 2020, 34, 643-649, doi:10.1038/s41433-019-0563-3.
  5. Pang, C.E.; Spaide, R.F.; Freund, K.B. Epiretinal proliferation seen in association with lamellar macular holes: a distinct clinical entity. Retina 2014, 34, 1513-1523, doi:10.1097/iae.0000000000000163.